# Sex-Specific Differences in LPS-Induced Rapid Myocardial Dysfunction

**DOI:** 10.3390/ijms26135963

**Published:** 2025-06-21

**Authors:** Brianna I. Harvey, Arris M. Yoniles, Andrea Monsivais, Jiayue Du, Lauren Zadorozny, Qing Yu, Meijing Wang

**Affiliations:** 1Center for Surgical Sciences, Department of Surgery, Indiana University School of Medicine, Indianapolis, IN 46202, USAayoniles@iu.edu (A.M.Y.); qyu@iu.edu (Q.Y.); 2Division of Cardiothoracic Surgery, Department of Surgery, Indiana University School of Medicine, Indianapolis, IN 46202, USA

**Keywords:** cardiac dysfunction, endotoxemia, gender difference, OXPHOS

## Abstract

Cardiac dysfunction is a severe complication of sepsis that significantly increases mortality in affected patients. Previous studies have shown better myocardial responses with preserved cardiac function in female animals compared to males following lipopolysaccharide (LPS)-induced sepsis. Our published findings have revealed that females exhibited less cardiac dysfunction than males when exposed to equivalent doses of tumor necrosis factor (TNF)α, which is markedly elevated in both heart tissue and serum following LPS. These raise the question of whether the observed sex differences in LPS-induced myocardial dysfunction are a direct effect of LPS or a secondary consequence mediated by inflammatory cytokines, like TNFα. In this study, we aimed to uncover sex differences in LPS-caused direct effects on cardiac function. To do so, isolated hearts from aged-matched adult male and female mice were subjected to LPS infusion using a Langendorff method. Left ventricular developed pressure (LVDP) was continuously recorded. The female estrous cycle was determined via vaginal smear. The oxidative phosphorylation (OXPHOS) pathway and estrogen receptors (ERs) were determined in heart tissue using Western blot. We found that males exhibited worse LV function than females following the infusion of LPS at 5.0 mg/kg body weight. However, no significant differences in cardiac function and expression of ERs were observed between female groups at different estrous stages. Interestingly, LV function returned to baseline after the initial depression of LVDP during the rapid response to LPS and then depressed again following the 50 min LPS infusion. Protein levels of OXPHOS were altered differently between male and female hearts after 50 min LPS infusion. Our data demonstrate that male hearts exhibit higher sensitivity to LPS-induced rapid cardiac dysfunction compared to females, although estrogen may have a minimal influence on LPS-induced rapid functional depression. Sex differences may exist in myocardial mitochondrial responses to direct LPS insult via the OXPHOS pathway.

## 1. Introduction

Lipopolysaccharide (LPS), a component of the outer membrane of Gram-negative bacteria, is a major trigger of endotoxemia. LPS can lead to sepsis—a life-threatening syndrome marked by organ dysfunction due to a dysregulated host response. A serious complication of sepsis is cardiac dysfunction, known as sepsis-induced cardiomyopathy, which significantly contributes to increased mortality in septic patients [1,2,3,4]. Understanding the molecular mechanisms driving cardiac dysfunction and damage is critical for developing effective therapeutic approaches to improve patient outcomes.

Accumulating evidence suggests sex-specific differences in the cardiovascular system response to injury. Of note, previous studies have demonstrated that female animals exhibit improved myocardial responses and better cardiac function compared to males following LPS challenge [5,6]. However, it remains unclear whether these sex differences are due to the direct effects of LPS or are mediated by LPS-induced inflammatory cytokines. In fact, our published study, along with others, has shown a significant increase in tumor necrosis factor (TNF)α levels in both heart tissue and serum after LPS challenge [5,7,8,9]. In addition, females experienced less cardiac dysfunction than males when exposed to an equivalent dose of TNFα [10,11]. This suggests that sex-related disparities in myocardial impairment could be attributable to indirect or secondary effects of LPS-induced inflammatory cytokines, such as TNFα. Therefore, in this study, to simulate endotoxemia and assess myocardial responses without the influence of confounding blood-borne factors, we employed the Langendorff perfusion technique. We aimed to investigate the direct effects of LPS on cardiac function by administering a coronary infusion of LPS to isolated male and female mouse hearts.

## 2. Results

### 2.1. LPS Directly Induces Cardiac Functional Depression

The lethal dose (LD50—the dose causing death in 50% of the mice) of LPS is 10 mg/kg body weight for C57BL/6 mice [12]. We first conducted dose-responsive experiments with LPS at 0, 2.5, 5.0, and 7.5 mg/kg body weight in isolated male mouse hearts (the infusion concentrations of LPS were 0, 31.25, 62.5, and 93.75 μg/mL, according to if a mouse had a total blood volume of about 80 mL/kg body weight [13]). Our data demonstrated that myocardial function was impaired by direct LPS infusion (Figure 1B), with a significant depression of left ventricular developed pressure (LVDP), and the maximal positive and negative values of the first derivative of pressure (+dP/dt and −dP/dt) in doses of LPS at 5 and 7.5 mg/kg body weight. However, the infusion of LPS at 2.5 mg/kg body weight did not cause myocardial dysfunction in male hearts. Of note, the 7.5 mg/kg dose of LPS resulted in a comparable level of cardiac functional depression to that observed with the 5 mg/kg LPS infusion (Figure 1B), in line with the previous report showing that both 5 and 10 mg/kg LPS challenges caused a substantial and comparable reduction in cardiac function in vivo [14]. In addition, we found that the myocardium responded to LPS as quickly as 5 min after a 5 mg/kg infusion of LPS (Figure 1A). Collectively, we did not increase the LPS dose further to 10 mg/kg and selected 5 mg/kg to investigate the direct effect of LPS on cardiac dysfunction in male versus female hearts in the experiments.

### 2.2. Male Hearts Experienced Greater Depression of Myocardial Function Compared to Female Hearts in Response to Infusion of LPS at 5 mg/kg Body Weight

Following exposure to a 5 mg/kg infusion of LPS, male myocardial function significantly decreased by about 50% in LVDP and −dP/dt and by 40% in dP/dt (Figure 2A), while female hearts exhibited a decrease of approximately 30% in LVDP and 20% in dP/dt and −dP/dt, highlighting a sex difference in LPS-induced acute myocardial functional depression. Additionally, LPS triggered a significantly faster cardiac response in male hearts (6.15 ± 1.22 min) compared to female myocardium (13.09 ± 1.92 min) (Figure 2B).

### 2.3. Effects of Estrous Cycle on LPS Caused Myocardial Dysfunction in Females and LPS’ Role in Modulating Expression of Estrogen Receptors in Male and Female Hearts

Estrogen, a sex hormone, plays a role in influencing cardiac function during injury. Female animals at different estrous stages have varying estrogen levels, which may impact myocardial responses to LPS. To explore this, we divided the female mice into two groups based on vaginal smear results (Figure 3A) as follows: proestrus/estrus females with high estrogen levels and metestrus/diestrus females with low estrogen levels. Interestingly, there was no significant difference in myocardial function between proestrus/estrus and metestrus/diestrus females following a 5 mg/kg infusion of LPS (Figure 3B). Additionally, considering that estrogen exerts its biological function through binding to estrogen receptors (ERs), we investigated whether the differential expression of the classical receptors ERα and ERβ occurs in the heart following LPS infusion. We found that there were no sex differences in the myocardial expression of ERα and ERβ without LPS stress (Figure 3(C1,C2,D1,D2)). In addition, LPS did not change the levels of ERα and ERβ in male hearts (Figure 3(C2,D2)). However, significantly reduced ERα and ERβ expression was observed in female hearts following LPS infusion (Figure 3(C2,D2)). Furthermore, the estrus cycle did not alter LPS-decreased levels of ERα and ERβ in female hearts (Figure 3(C3,D3)). These data suggest that estrogen and/or estrogen signaling may not influence the direct effects of LPS on myocardial function.

### 2.4. Effects of Longer LPS Infusion on Isolated Male and Female Heart Function and on Myocardial Expression of Toll-like Receptor 4

Notably, when the duration of the LPS infusion (5 mg/kg body weight) was extended to 50 min (50′), we observed that LV function returned to normal after the initial rapid depression/first response to the LPS (<20′ infusion). However, LV function significantly decreased again in male hearts following the 50′-LPS infusion (Figure 4A,B). In contrast, although female hearts showed a similar trend, no statistically significant difference was observed during the 50′-LPS infusion (Figure 4C). This suggests that the female myocardium may be more tolerant to direct LPS-induced injury.

LPS binds to Toll-like receptor 4 (TLR4), initiating the LPS/TLR4 signaling pathway to exert its biological effects. Therefore, we assessed myocardial TLR4 expression in male and female hearts in response to LPS infusion. We found that LPS infusion did not change TLR4 levels in either male or female hearts (Figure 4D).

### 2.5. Direct Effects of LPS on Myocardial Mitochondrial OXPHOS Between Male and Female Hearts

LPS has been shown to impair mitochondrial metabolism in vivo via the OXPHOS pathway in the heart [15]. To dissect the molecular mechanisms underlying the sex differences in LPS-induced cardiac dysfunction, we examined the myocardial protein levels of mitochondrial OXPHOS. Increased levels of ATP5A, UQCRC2, and NDUF8 were observed in male hearts following 50′-LPS infusion (Figure 5A,B), whereas 50′-LPS infusion significantly reduced UQCRC2 and MTCO1 levels in female hearts (Figure 5A,B), indicating sex-specific differences in myocardial mitochondrial responses to direct LPS insult.

## 3. Discussion

Sex differences in myocardial responses have been reported following LPS challenge in vivo [5,6]. However, our published studies and others have indicated that inflammatory cytokines, including TNFα, substantially increased following LPS injection [5,7,8,9]. Additionally, male hearts had more severe cardiac contractile depression than female hearts when exposed to TNFα [10,11], raising the question as to whether sex differences in the myocardial response to LPS in vivo are due to LPS-increased inflammatory cytokines in the circulation, particularly TNFα. In this study, we investigated the direct effects of LPS on myocardial function between male and female hearts using the Langendorff approach, which eliminates the influence of confounding blood-borne factors. Our results showed that when exposed to the same dose of LPS (1) male hearts experienced greater myocardial functional depression compared to female hearts; (2) no significant differences in cardiac dysfunction and myocardial ER expression were observed between proestrus/estrus and metestrus/diestrus females; and (3) distinct differences in OXPHOS responses were noticed between male and female hearts.

LPS is a pathogen-associated molecule and functions as TLR4 agonists to initiate the LPS/TLR4 signaling pathway. All cardiac cell types including cardiomyocytes express TLR4 [16]. However, most studies focus on evaluating LPS/TLR4 signaling in inflammatory responses and reactive oxygen species (ROS) production [17,18], thus subsequently regulating myocardial function during endotoxemia. In fact, LPS decreased the sarcoplasmic reticulum Ca^2+^ content, the systolic Ca^2+^ transient, and cardiomyocyte contraction, all of which were reversed by using the TLR4-specific inhibitor or deletion of TLR4 [19]. While LPS-induced cardiomyocyte impairment requires TLR4 on immune cells not on cardiomyocytes, as LPS does not cause myocyte shortening in mice with TLR4-deficient leukocytes [20], other studies have challenged this view by demonstrating that LPS-depressed cardiac function was not observed in mice with TLR4KO in cardiomyocytes [21]. Such a discrepancy may be due to the in vivo LPS treatment, which induces systematic responses and changes in blood-borne factors to complicate outcomes. In the current study, we explored the direct effects of LPS on impairing myocardial function in isolated mouse hearts, excluding other confounding variables. Our previous work has also shown that the direct infusion of LPS into isolated rat hearts via coronary delivery led to a progressive depression of LV contractile function [22]. Interestingly, in this study, we found that the myocardial responses to LPS appeared time-dependent, with a cycle of rapid depression, returning to normal, and decreasing in LV function again along with continuing LPS infusion. This result suggests that a low dose/amount of LPS (the first depression) likely induces the defense mechanisms to prevent the heart from further injury, leading to LV function returning to normal. Our finding confirmed the possibility to use LPS for eliciting the preconditioning mechanism in the heart, aligning with the previous studies in which LPS induced preconditioning to improve postischemic cardiac function and reduce infarct size against myocardial infarction [23,24,25].

Of note, sex differences have been observed in the prognosis of sepsis. Clinically, epidemiological studies consistently indicate that severe sepsis with developed organ dysfunction/failure, including cardiac dysfunction, often occur in male patients [26,27,28,29]. In line with this, myocardial functional depression was significantly less noticeable in female mice compared to males following LPS (3 mg/kg) injection [5]. Additionally, endotoxic shock remained in male rats with depressed LV function and decreased stork work 24 h after LPS challenge, whereas cardiac function was significantly improved in 24 h LPS-treated female rats [6]. In our study, by excluding the confounding systematic and blood-borne influences, sex differences in cardiac dysfunction were further noticed following the direct infusion of LPS, with female mouse hearts more resistant to an equal amount of LPS compared to male hearts. Accumulated evidence has reported that the sex differences in cardiac function and inflammatory responses appears attributable to sex hormones—estrogen and its downstream signals during endotoxemia [8]. Using estrogen or selective agonists of ERs protected male rats against LPS-induced damage, while the depletion of estrogen by ovariectomy worsened the detrimental effects of LPS on female rats [8,30]. To this end, we investigated whether estrogen played a role in LPS-induced rapid LV dysfunction in female mice with various estrogen levels in the present study. However, we did not see significant differences in the LPS-depressed cardiac function between the proestrus/estrus and metestrus/diestrus female hearts, implying that fluctuations in estrogen may have little influence on the acute myocardial response to LPS. Notably, LPS significantly reduced the myocardial expression of ERα and ERβ in female hearts but not in male hearts. Interestingly, endogenous estrogen levels did not appear to impact the levels of ERs. These findings suggest potentially changed estrogen signaling in female hearts compared to the male myocardium during endotoxemia, warranting further investigation.

Next, we asked a question as to how differences in myocardial dysfunction following the direct infusion of LPS exist between male and female mouse hearts. Indeed, accumulating evidence has shown sex differences in sarcomeric protein composition and calcium-handling function, contributing to stronger and faster contractions in male cardiomyocytes compared to female cells [31,32]. These sex-specific differences in cardiac structure and function may influence the heart’s response to LPS infusion. However, we did not observe differences in TLR4 levels between male and female hearts in response to LPS, suggesting that the observed sex disparities in cardiac function are not due to differential TLR4 signaling following LPS infusion. Nevertheless, the Four Core Genotypes mouse model can be employed in future studies to investigate the relative contributions of sex chromosomes and gonadal sex to the molecular mechanisms underlying sex differences during sepsis.

On the other hand, compelling evidence has suggested a key role of mitochondrial dysfunction and metabolic shutdown in the pathogenesis of sepsis-induced organ dysfunction/failure [33,34]. Notably, emerging studies raise the possibility that cardiac mitochondria are more vulnerable to sepsis-induced impairment than those from kidneys, the liver, and skeletal muscles [35,36,37]. In fact, sex differences in mitochondrial function have been reported, with less mitochondrial ROS production and oxidative damage in female hearts [38,39,40]. Our previous study has also shown better mitochondrial metabolic function in cardiomyocytes from female mice compared to those from males when exposed to inflammatory mediators—TNFα or oxidative stress—H_2_O_2_ [41]. In the present study, we observed a significant increase in mitochondrial respiratory chain proteins complex I-NDUFB8, complex III-UQCRC2, and complex V-ATP5A in male mouse hearts subjected to LPS, while there was a marked decrease in complex III-UQCRC2 and complex IV-MTCO1 in the LPS-infused female mouse hearts. The activity of mitochondrial complex I determines the overall electron flux through the respiratory chain (complexes I-IV), significantly contributing to ROS production. Elevated levels of complex I have been observed in the male brain [42,43], which may help explain the higher oxidative stress reported in males [44]. Our findings of increased complex I content are consistent with these observations, suggesting that sex differences likely exist in mitochondrial respiratory function, with female hearts experiencing less oxidative stress than male hearts following 50 min of LPS infusion. In addition, our results on sex differences in OXPHOS enzyme content may also suggest differing adaptive capacities to altered metabolic conditions between male and female hearts. Indeed, stressed females exhibited a more protective and resilient phenotype of major mitochondrial respiratory complexes compared to males [45]. Furthermore, enhanced metabolic adaptability has been reported in female skeletal muscle under changed energy conditions [46].

Limitations: In this study, we utilized a dose of LPS at 5 mg/kg body weight to investigate sex differences in cardiac dysfunction. It is unknown whether these sex disparities persist with higher doses of LPS. Additionally, the longest duration of LPS infusion in the current study was 50 min, and it remains unclear what effects might occur with an infusion lasting longer than 50 min. While vaginal smear can serve as a useful tool for training medical students in basic techniques in the current study, measuring estradiol levels in serum provides more accurate results for determining estrogen level in circulation and should be used in future research. Furthermore, future studies incorporating exogenous estrogen or ER modulators (both agonists and antagonists) into the perfusate should be undertaken to investigate the role of exogenous estrogen and membrane-associated ERs in acute/rapid cardioprotection during LPS infusion.

## 4. Materials and Methods

### 4.1. Animals

Male and female C57BL/6J mice (9–12 weeks) were purchased from Jackson Laboratories (Bar Harbor, ME, USA). All mice were acclimated for >5 days with a standard diet feeding prior to the experiments. Age-matched male and female mice at 12–16 weeks old were used for the experiments. The animal protocol was reviewed and approved by the Institutional Animal Care and Use Committee of Indiana University. All animals received humane care in compliance with the Guide for the Care and Use of Laboratory Animals (NIH Pub. No. 85-23, revised 1996).

### 4.2. Assessment of Cardiac Function Using Langendorff Method

A modified isovolumetric Langendorff technique was employed to evaluate intrinsic cardiac contractility without or with LPS infusion. Isolated mouse hearts were subjected to the Langendorff technique, as previously described [10,11,22,47,48]. Briefly, mice were anesthetized with isoflurane and heparinized (100 IU i.p.). Hearts were then rapidly excised into ice-cold Krebs–Henseleit (K-H) buffer. The aorta was annulated under a dissection microscope and the heart was perfused in the isovolumetric Langendorff mode (70 mmHg) with oxygenated (95% O_2_-5% CO_2_) K-H solution (37 °C). The hearts were paced at 420–460 bpm/min during the whole experimental time. Data were continuously recorded using a PowerLab 8 preamplifier/digitizer (ADInstruments Inc., Colorado Springs, CO, USA). The LVDP was measured, +dP/dt and −dP/dt) were calculated using PowerLab software (LabChart 8).

### 4.3. Experimental Protocols

A dose–response study was performed in isolated adult male mouse hearts with LPS (from *Escherichia coli* O127:B8, L3129, MilliporeSigma, Burlington, MA, USA) at 0, 2.5, 5.0, and 7.5 mg/kg body weight. The infusion concentrations of LPS were 0, 31.25, 62.5, and 93.75 μg/mL, based on the fact that the total blood volume of a mouse is approximately 80 mL/kg [13]. Isolated mouse hearts were subjected to an infusion protocol consisting of a minimum 20 min equilibration period, followed by LPS infusion through the aortic root. The experimental workflow is shown in Figure 6. The concentration was determined with 5 mg/kg body weight (62.5 μg/mL) for all experiments in female hearts and the study in male hearts with a 50 min infusion period. The hearts were harvested at the time point when a significant depression of LVDP occurred, with a >20% drop following LPS infusion in male hearts and a >10% drop in female hearts and after a 50 min infusion period. Heart tissue was snap-frozen in liquid nitrogen and stored at −80 °C.

### 4.4. Vaginal Smear

After the heart was removed, 50 μL of sterile PBS was gently introduced into the vaginal canal of female mice using a pipette. The pipette was aspirated and dispensed several times to wash the vaginal canal and collect an adequate number of cells for a single sample. The fluid was then placed on a glass slide. After allowing the smear to air dry for 10 min, it was examined under light microscopy, and photomicrographs were taken. The images were analyzed by at least three researchers to determine the cell type present. Based on vaginal cytology, four estrus stages were identified [49], namely proestrus, characterized by clusters of well-formed nucleated epithelial cells; estrus, marked by clusters of cornified squamous epithelial cells; metestrus, showing fragmented cornified squamous epithelial cells with small leukocytes; and diestrus, dominated by small leukocytes with a few cornified squamous epithelial cells.

### 4.5. Western Blotting

The heart tissues were lysed in cold RIPA buffer containing Halt protease and phosphatase inhibitor cocktail (ThermoFisher Scientific, Waltham, MA, USA). The protein extracts (10–20 μg) from heart tissue were subjected to electrophoresis on a 4–15% Criterion TGX Precast midi protein gel (Bio-Rad, Hercules, CA, USA) and transferred to a nitrocellulose membrane. The membranes were incubated with the following primary antibodies, respectively: OXPHOS antibody cocktail (complex I-NDUFB8, complex II-SDHB, complex IV-MTCO1, complex III-UQCRC2, and complex V-ATP5A) (ThermoFisher Scientific), ERα (SC-543), ERβ (SC-8974), TLR4 (SC-293072) (Santa Cruz, Dallas, TX, USA), and GAPDH (#5174) (Cell Signaling Technology, Beverly, MA, USA), followed by a HRP-conjugated secondary antibody. The images were detected by a ChemiDoc system (Bio-Rad). Image J software (Version 1.51, NIH) was utilized for immunoblotting band density measurement.

### 4.6. Statistical Analysis

All reported results were means ± SEM, and each dot was an individual heart measurement. Data were checked for variables using the Shapiro–Wilk normality test and then evaluated using an unpaired t-test or analysis of variance (ANOVA) followed by a multiple comparisons test. Differences were considered statistically significant when *p* < 0.05. All statistical analyses were performed using GraphPad Prism (Version 10.5.0, GraphPad, La Jolla, CA, USA).

## 5. Conclusions

In summary, the present study clearly demonstrates that sex differences exist in cardiac dysfunction following the direct infusion of LPS, with significantly depressed LV contractile function in male hearts compared to female hearts. Given the comparable levels of LPS-induced cardiac dysfunction between the proestrus/estrus and metestrus/diestrus groups, estrogen itself may have little effect on female hearts in response to direct LPS infusion. However, the current findings cannot fully rule out estrogen’s role in regulating the female myocardial response to LPS infusion for the two following reasons: (1) it remains unclear whether estrogen levels varied in the hearts of female mice at different estrus stages and (2) the isolated hearts were perfused with the same buffer without estrogen. Further research is needed to explore the mechanisms behind sex-mediated LPS-induced cardiac dysfunction, particularly the mitochondrial factors driving differences in the myocardium. Understanding these important unknowns could pave the way for the development of novel therapies for septic cardiomyopathy.

## Figures and Tables

**Figure 1 ijms-26-05963-f001:**
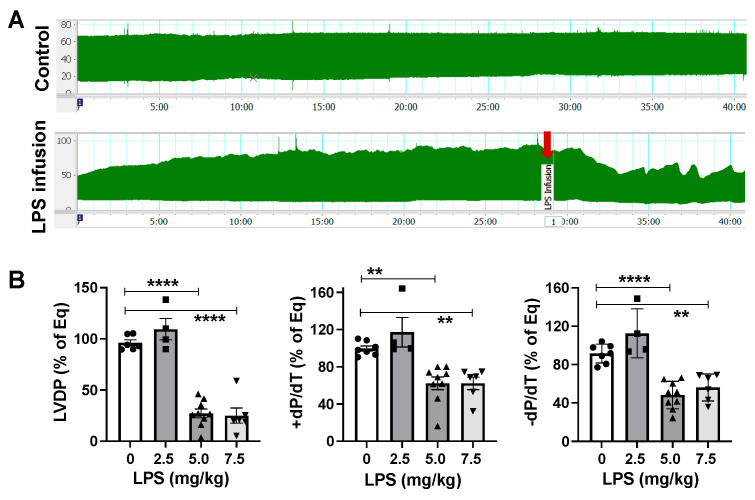
(**A**) LVDP (left ventricular developed pressure) trace in male hearts without (control) or with LPS infusion (5.0 mg/kg). (**B**) Dose–response study of LPS infusion in isolated male mouse hearts using Langendorff technique. Dots represent individual mouse heart function. Mean +/− SEM, ** *p* < 0.01, **** *p* < 0.0001.

**Figure 2 ijms-26-05963-f002:**
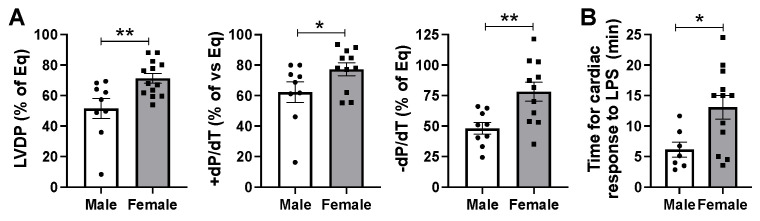
(**A**) Heart function in male and female hearts after infusion of 5.0 mg/kg LPS at rapid response/initial depression of LV function. (**B**) Time required for rapid response/initial depression of LV function in male and female hearts after infusion of 5.0 mg/kg LPS. Dots represent individual mouse heart values. Mean +/− SEM, * *p* < 0.05, ** *p* < 0.01.

**Figure 3 ijms-26-05963-f003:**
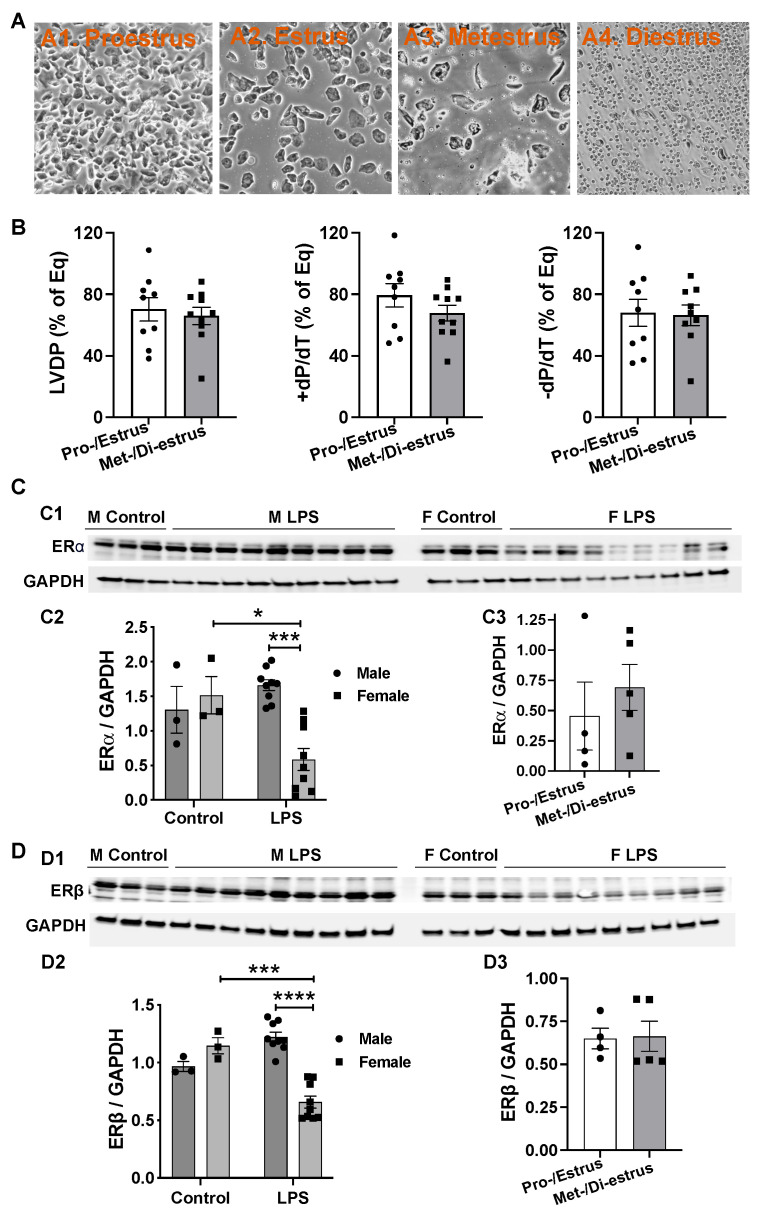
(**A**) Estrus cycle in female mice by vaginal smear. High-estrogen stages: (**A1**) proestrus with clusters of well-formed nucleated epithelial cells; (**A2**) estrus with clusters of cornified squamous epithelial cells. Low-estrogen stages: (**A3**) metestrus shows fragmented cornified squamous epithelial cells accompanied by small leukocytes; (**A4**) diestrus dominated by small leukocytes with few cornified squamous epithelial cells. (**B**) Cardiac function in female mouse hearts during proestrus/estrus (high estrogen) versus metestrus/diestrus (low estrogen) phases following LPS infusion. (**C**) ERα expression in male and female hearts without LPS infusion. (**C1**) Western blot images of ERα and GAPDH across groups; (**C2**) quantification of ERα band intensity normalized to GAPDH; (**C3**) comparison of relative ERα levels (normalized to GAPDH) between high-estrogen (proestrus/estrus) and low-estrogen (metestrus/diestrus) female hearts following LPS infusion. (**D**) ERβ expression in male and female hearts +/− LPS infusion. (**D1**) Western blot images of ERβ and GAPDH among groups; (**D2**) quantification of ERβ band intensity normalized to GAPDH; (**D3**) comparison of relative ERβ levels (normalized to GAPDH) between high-estrogen (proestrus/estrus) and low-estrogen (metestrus/diestrus) female hearts after LPS infusion. Dots represent individual mouse heart functions. Mean +/− SEM; two-way ANOVA in C2 and D2, *t*-test in others; * *p* < 0.05, *** *p* < 0.001, **** *p* < 0.0001.

**Figure 4 ijms-26-05963-f004:**
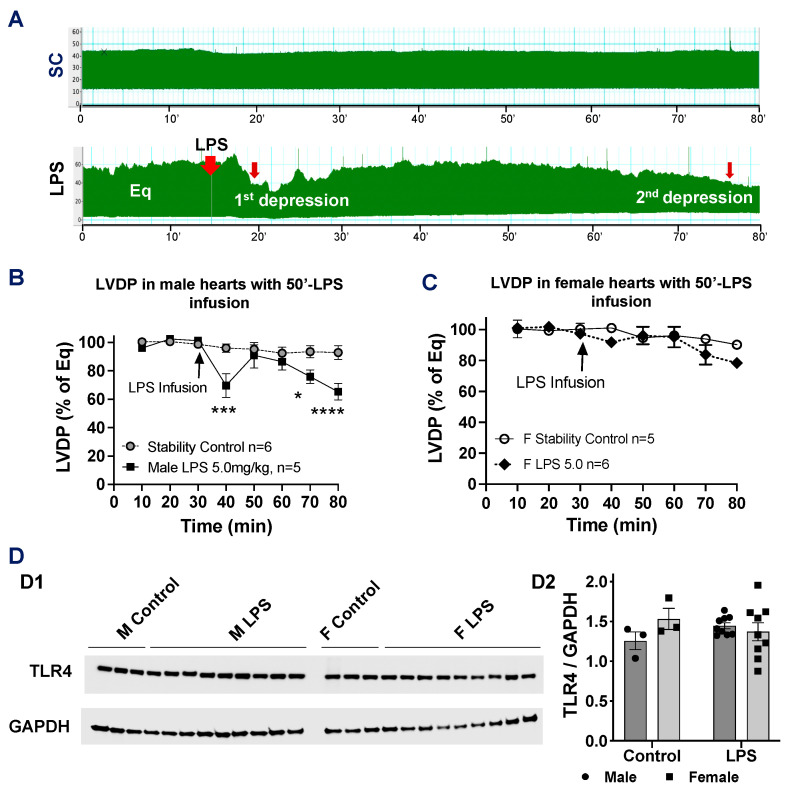
(**A**) Representative LVDP trace of male stability control (SC) with 80 min of perfusate perfusion and male heart with 50 min of LPS infusion (5.0 mg/kg). Changes in LVDP in SC and LPS-infused male hearts (**B**) and female hearts (**C**). (**D**) TLR4 expression in LPS-infused male and female hearts compared to their untreated counterparts. (**D1**) Western blot images of TLR4 and GAPDH across groups; (**D2**) quantification of TLR4 band intensity normalized to GAPDH. Mean +/− SEM; two-way ANOVA; * *p* < 0.05, *** *p* < 0.001, **** *p* < 0.0001.

**Figure 5 ijms-26-05963-f005:**
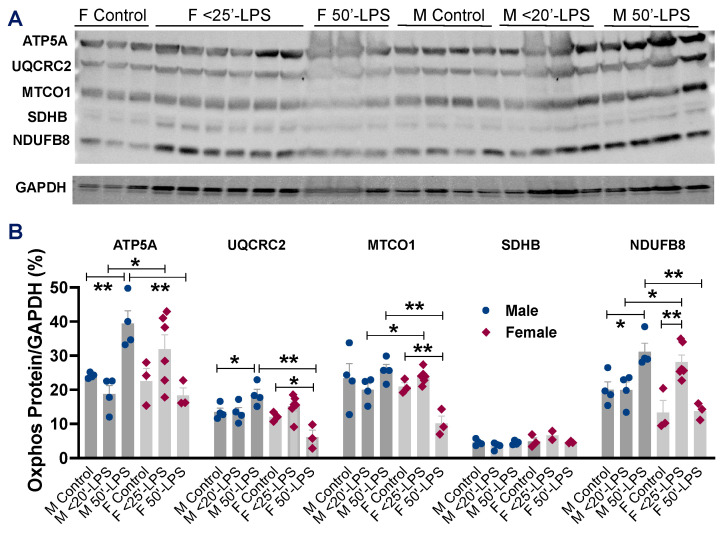
(**A**) Immunoblots of five OXPHOS protein levels in female (F) and male (M) hearts with +/− <25 min or 50 min LPS infusion. (**B**) Quantification of immunoblotting band intensity of ATP5A-complex V, UQCRC2-complex III, MTCO1-complex IV, SDHB-complex II, and NDUFB8-complex I compared to GAPDH, respectively. Mean +/− SEM; * *p* < 0.05, ** *p* < 0.01. One-way ANOVA with multiple comparisons within the same sex group and a *t*-test between M and F at the same dose of LPS infusion.

**Figure 6 ijms-26-05963-f006:**
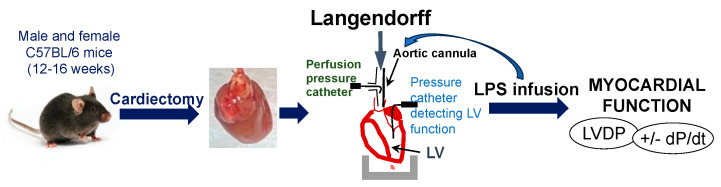
The scheme shows the experimental workflow.

## Data Availability

Data is contained within the article.

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
