# Peer review of "Sex-Specific Differences in LPS-Induced Rapid Myocardial Dysfunction"

_ijms, 2025, doi:10.3390/ijms26135963_

Round 1
Reviewer 1 Report
Comments and Suggestions for Authors
The author aimed to confirm that sex differences may exist in myocardial mitochondrial responses to direct LPS insult via the OXPHOS pathway. The work use the Langendorff technique, providing novel insights into sepsis-related cardiac complications.
Here are my comments:
1) The authors refered that LD50 is ~10 mg/kg, but most experiment limited to 7.5 mg/kg, which needs to be explained.
2)Figure 3 could be more clear.
3)Some error bars are missing in Figure 4.
4)Biological implications of OXPHOS could be discussed to provide broader interest.
5) Line 145 Ca2+ should be Ca2+
4)In discussion part, the sexual effect is main defined by estrogen, which is somehow not included in the research especially in vivo exeperiment.
Reviewer 2 Report
Comments and Suggestions for Authors
The relevance of this study is determined by new data regarding the presence or absence of gender-specific features in mitochondrial reactions induced by endotoxin exposure, which could be useful in terms of improving the efficacy of septic shock therapy in men and women. Strengths of the study: The study presents comprehensive comparative analysis of heart function up to the subcellular level, in male and female models under direct endotoxin exposure. This includes assessing the direct effect of LPS on cardiac functional depression and mitochondrial metabolism in the main and control groups, evaluation of heart function and the time required for a rapid response/initial left ventricular (LV) function depression, effects of lipopolysaccharide infusion duration in male and female hearts, as well as investigation of the influence of the estrous cycle on septic myocardial dysfunction in females. The obtained data may serve as a basis for developing effective therapeutic programs tailored separately for male and female populations in the future. A non-critical question that could increase interest in this manuscript: Was the nitric oxide (NO) content in the myocardium and/or the end products of NO oxidation (nitrate and nitrite, NOx) measured? This could potentially help characterize different inflammatory stress responses in male and female hearts. Conclusion: The manuscript does not require revisions or additional peer review and can be published in its current form.
Reviewer 3 Report
Comments and Suggestions for Authors
General:
It was reported that female animals displayed better cardiac function compared to males following LPS challenge; this study was attempt to examine the direct effects of LPS infusion on cardiac dysfunction using isolated male and female murine heart using a Langendorff procedure. It was observed that the male heart functions depressed quickly and more severely compared to the females; furthermore, the males but not the females depressed again after 50 min infusion. It was described that different estrous stages had no impact on the dysfunction experienced by the female hearts. However, protein levels of OXPHOS were altered differently between male and female hearts after 50-min LPS infusion. Overall, the new findings are interesting and noteworthy, but the data are somewhat limited and insufficient to justify the conclusions. Acquiring additional experimental data are suggested to improve the validity of the study.
Specifies:
- The role of estrogen on the impaired cardiac function was not sufficiently clarified by the experimental setting simply by using female hearts of different estrous stages; inclusion of estrogen in the perfusate (in addition to LPS) might be helpful. Also assess if differential expression of estrogen receptors on the heart may be useful as well.
- Altered expressed protein levels of OXPHOS were described between male and female hearts after 50-min LPS infusion; however, it was unknown if these specific changes would contribute to differential production of oxy-radicals. Additional experiments should be able to address this issue.
Round 2
Reviewer 3 Report
Comments and Suggestions for Authors
A duplication of Fig. 3A and 3B is noted on pg 4, 5.
Comments on the Quality of English Language
A minor point: Ln 229. the sentence can be shortened as following: Next, we asked whether differences in myocardial dysfunction following the direct infusion of LPS exist between.............hearts.